



# Global Spectroscopic Survey of Cloud Thermodynamic Phase at High Spatial Resolution, 2005-2015

David R. Thompson[1], Brian H. Kahn[1], Robert O. Green[1], Steve A. Chien[1], Elizabeth M. Middleton[2], and Daniel Q. Tran[1]

[1]Jet Propulsion Laboratory, California Institute of Technology, Pasadena, CA, USA
[2]Goddard Space Flight Center, Greenbelt, MD, USA

*Correspondence to:* David R. Thompson (david.r.thompson@jpl.nasa.gov)

**Abstract.** The distribution of ice, liquid, and mixed phase clouds is important for Earth's planetary radiation budget, impacting cloud optical properties, evolution, and solar reflectivity. Most remote orbital thermodynamic phase measurements observe kilometer scales and are insensitive to mixed phases. This under-constrains important processes having outsize radiative forcing impact, such as spatial partitioning in mixed phase clouds. To date, the fine spatial structure of cloud phase has not been

measured at global scales. Imaging spectroscopy of reflected solar energy from 1.4 - 1.8 $\mu$m can address this gap: it directly measures ice and water absorption, a robust indicator of cloud top thermodynamic phase, with spatial resolution of tens to hundreds of meters. We report the first such global high spatial resolution survey based on data from 2005-2015 acquired by the Hyperion imaging spectrometer onboard NASA's Earth Observer 1 (EO-1) spacecraft. Seasonal and latitudinal trends corroborate observations by the Atmospheric Infrared Sounder (AIRS). For extra tropical cloud systems, just 25% of variance

observed at GCM grid scales of 100 km was related to irreducible measurement error, while 75% was explained by spatial correlations possible at finer resolutions.

## 1 Introduction

The distribution of ice, liquid, and mixed phase clouds is important for Earth's climate and planetary radiation budget (Chylek

et al., 2006; Martins et al., 2011). Cloud thermodynamic phase affects radiative forcing by modulating absorption of incoming solar radiation, particle evolution, and lifetime (Ehrlich et al., 2008; Tan and Storelvmo, 2016). Previous satellite observational studies have shown that clouds are shifting poleward in the Northern and Southern Hemisphere extratropical storm tracks (Bender et al., 2012; Marvel et al., 2015; Norris et al., 2016). Within these shifting storm tracks, climate model experiments with forcing from increased $CO_2$ have shown losses of cloud ice phase and gains of cloud liquid phase (Ceppi and Hartmann,

2015; McCoy et al., 2015). This makes cloud phase an important property for accurate and continuous monitoring. Moreover, recent studies indicate that spatial partitioning of ice and liquid particles within clouds has outsized influence on climate, affecting Global Climate Model (GCM) predictions of future warming by over 1 degree Celsius (Tan et al., 2016). However,





current observing systems cannot reduce this uncertainty; they are unable to resolve differences at critical sub-100 m scales, or are insensitive to mixed phase clouds altogether.

Imaging reflectance spectroscopy from 1.4 - 1.8 $\mu$m could address this gap. Prior work used this spectral interval to measure cloud phase based on the optical absorption properties of liquid and ice (Pilewskie and Twomey, 1987). The fraction of the total

path absorption due to liquid (the Liquid Thickness Fraction, or LTF) had high sensitivity to pure and mixed phases (Thompson et al., 2016). Liquid and ice show highly diagnostic absorption shapes which are robust to potential confounding effects such as surface reflectance, observation geometry, and mismatch in particle modeling assumptions. Remote imaging spectrometers measure these spectra from cloud tops at millions of spatial locations, resolving cloud phase at tens of meter scales (Thompson et al., 2016). This could constrain characteristic spatial lengths of these processes. However, global spectroscopic datasets have

not yet been analyzed in this way.

Here we report a global spectroscopic survey of cloud phase from 2005-2015 based on data from the Hyperion imaging spectrometer instrument onboard NASA's Earth Observer 1 (EO-1) spacecraft. We show seasonal and latitudinal changes of cloud thermodynamic phase, comparing them to measurements by NASA's Atmospheric Infrared Sounder, AIRS (Kahn et al., 2014). Hyperion also provides two novel contributions beyond prior records. First, its full spectrum fitting discriminates mixed

phases by directly measuring the relative contributions of physical cloud top liquid and ice absorption. Second, Hyperion measures phase at horizontal scales of 30 m. These properties allow the first rigorous characterization of the spatial scales governing cloud top thermodynamic phase. The study lays the groundwork for future orbital imaging spectrometers (Mouroulis et al., 2016), that can monitor cloud characteristics when cloud cover precludes their primary mission. Imaging spectroscopy investigations typically treat clouds as contamination, when in fact cloudy data can be exploited to dramatically increase these

instruments' useful data yield.

## 2 Method

The Hyperion imaging spectrometer operated on the sun synchronous EO-1 spacecraft for over a decade prior to decommissioning in 2017. Hyperion measured reflected solar energy from approximately 400-2450 nm with approximately 10 nm spectral sampling. It performed targeted acquisitions for specific regions of interest, with occasional pointing off nadir. Most

targets were on land, with a high concentration in the mid-latitude northern hemisphere. There was sparser coverage of extreme latitudes and oceans, but several island targets offered a view into cloud systems over ocean (Figure 1). Many targets of interest were revisited multiple times during the mission. Each targeted acquisition had a Ground Sampling Distance (GSD) of approximately 30 m over a cross-track width of approximately 7.5 km, and a typical along-track distance of approximately 120 km. The Jet Propulsion Laboratory (JPL) Hyperion archive included most normal science acquisitions from 2005 through

2015, with over $4.8 \times 10^4$ scenes and $3.7 \times 10^{10}$ distinct spectra.





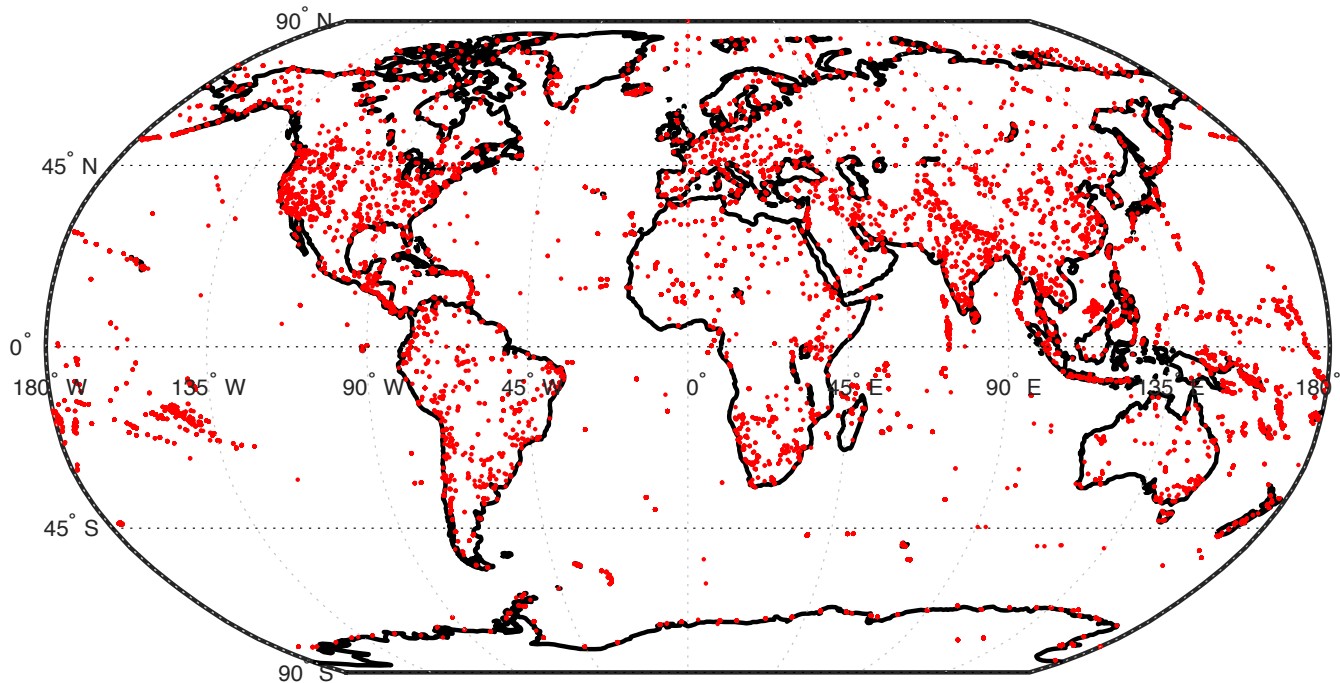

**Figure 1.** The JPL Hyperion archive comprised over 45,000 scenes, represented here by red dots. The majority were over land.

## 2.1 Phase Estimation

The archive was stored as standard calibrated unorthorectified data. We first applied the cloud phase retrieval algorithm of Thompson et al. (2016), validated in the prior work by coincident remote and in-situ aircraft measurements. We defined the TOA reflectance, $\rho$:

$$\rho(\lambda) = \pi L(\lambda) F(\lambda)^{-1} \, cos(\theta)^{-1} \tag{1}$$

where $\lambda$ was the wavelength, $L$ was the wavelength-dependent radiance at sensor, $F$ was the extraterrestrial solar irradiance, and $\theta$ is the solar zenith angle. Over short intervals we modeled $\rho$ by a linear continuum with an offset $a$ and slope $b$, attenuated by one or more Beer-Lambert absorbers $j$. Each absorber had a bulk absorption coefficient $k_j$ and a nonnegative thickness $u_j$:

$$\rho(\lambda) = (a + \lambda b) \, \exp[-\sum_j k_j(\lambda) u_j] \quad \text{for} \quad u_j \geq 0 \tag{2}$$

10    We modeled three absorbers: atmospheric gases, including water vapor; liquid water; and ice water. As in Thompson et al. (2015, 2016) we applied a logarithmic transformation resulting in a nonnegative least squares problem (Lawson and Hanson, 1974):

$$
\begin{aligned}
-log(\rho(\lambda)) \quad &\approx \quad l + \lambda m - \lambda n + \sum_j k_j(\lambda) u_j \\
&\text{for} \quad \{l, m, n, u_j \forall j\} \geq 0
\end{aligned}
\tag{3}
$$





Here $m$ and $n$ represented positive and negative ramp functions, allowing a log-linear continuum with upward or downward slope. The nonnegative least squares solution provided stable global solutions without sensitivity to initialization.

We calculated the $H_2O$ vapor absorption coefficients using the HITRAN 2012 line list (Rothman et al., 2013), via the Oxford Reference Forward Model (Dudhia, 2014). We initialized the vapor abundance using a band ratio retrieval as in Thompson et al.

(2015), and used this to calculate an "effective" absorption coefficient of band-aggregated vapor lines for use in the least squares retrieval. Other atmospheric gases did not significantly impact the shape of vapor absorption features. We calculated liquid and ice absorption coefficients using the complex index of refraction measured by Kou et al. (1993). These bulk absorption spectra were generally independent of particle scattering and did not relate directly to particle size. However, combined with a free continuum, they fit the observed spectra of opaque clouds over short spectral intervals without precise knowledge of particle

properties (Thompson et al., 2016).

In summary, following Equation 3 we modeled the entire interval from $1.4 - 1.8\ \mu$m with five free parameters: a continuum offset $l$; a slope, represented by a single degree of freedom in the variables $m$ and $n$; and the vapor, ice and liquid thicknesses $u_j$. These thicknesses represented the equivalent length of the optical path through a homogeneous volume, as in the Equivalent Water Thickness (Gao and Goetz, 1995). Here as in prior work, we separated the Equivalent Water Thickness due to Ice,

$EWT_{ice}$ from that of liquid, $EWT_{liquid}$. As in Thompson et al. (2016), we defined the Liquid Thickness Fraction (LTF) as:

$$LTF = \frac{EWT_{liquid}}{EWT_{liquid} + EWT_{ice}} \tag{4}$$

This value is an effective proxy for thermodynamic phase.

## 2.2 Cloud Detection

We calculated the LTF of locations flagged by the Hyperion cloud detection algorithm of Griffin et al. (2003). In the Griffin et

al. approach, a decision tree of threshold tests sorted spectra into different cloud types (including low and high clouds) as well as different land cover types (including snow, open terrain surfaces, and vegetation). This favored opaque clouds and generally ignored ambiguous translucent clouds that would be difficult to distinguish from land. The Griffin et al. algorithm defines tests based on the top of atmosphere reflectance $\rho$ and three intermediate quantities, the Normalized Difference Snow Index (NDSI), the Desert Sand Index (DSI), and the Vegetation Index (VI):

$$NDSI = [\rho(0.55) - \rho(1.65)] / [\rho(0.55) + \rho(1.65)] \tag{5}$$
$$DSI = [\rho(0.86) - \rho(1.65)] / [\rho(0.86) + \rho(1.65)] \tag{6}$$
$$VI = \rho(0.66)/\rho(0.86) \tag{7}$$

The method was originally formulated as a flowchart but is tantamount to an ordered sequence of tests (Table 1). It evaluated each spectrum independently, applying appropriate thresholds depending on whether the scene was over ocean or land. It

ascribed a classification based on the first test that evaluated to true. Random manual validation of selected scenes demonstrated adequate performance, outside the brightest glacial scenes in Antarctica.





|  | **Criterion** | | |
| **Test** | **Over Ocean** | **Over Land** | **If True** |
| --- | --- | --- | --- |
| 1. High cloud | $\rho(1.38) > 0.1$ | $\rho(1.38) > 0.1$ | Cloud |
| 2. Land / water | $\rho(0.66) < 0.15$ | $\rho(0.66) < 0.15$ | Clear |
| 3. Vegetation | VI < 0.6 | VI < 0.7 | Clear |
| 4. Desert sand | DSI < 0.01 | DSI < 0.05 | Clear |
| 5. Low cloud | -0.2 < NDSI < 0.2 | 0 < NDSI < 0.2 | Cloud |
| 6. Snow or ice | 0.6 < NDSI | 0.6 < NDSI | Clear |
| 7. Snow or ice | $\rho(1.25) < 0.35$ | $\rho(1.25) < 0.35$ | Clear |
| 8. Snow or ice | $0.1 < \rho(1.38)$ | $0.1 < \rho(1.38)$ | Clear |

**Table 1.** Hyperion Cloud Detection algorithm Griffin et al. (2003). We test each criterion in sequence, starting from the top, and ascribe a classification based on the first test that evaluates to true.

## 2.3 Zonal Statistics

After calculating LTF maps over all cloud areas, we aggregated counts of liquid and ice clouds. LTF was a continuous-valued quantity so the maps revealed both pure and mixed phases. We binned LTF values in 10% graduations but also calculated binary classification using the dominant absorber. This facilitated comparisons with historical datasets of hard categorical clas-

5 sifications. We then analyzed zonal trends, estimating confidence intervals with nonparametric bootstrap variance estimation (Wasserman, 2006) that resampled the dataset 10,000 times with replacement.

## 2.4 Goodness of Fit

Evaluating the model fit to the spectrum demanded special care, since the global catalogue included many observing geometries, terrain surface types, and potential variability in instrument calibration over the decadal record. To account for this, we

10 estimated the noise level independently for each integration timestep and each spectral channel. We used the common method of pairwise differences between spectra at neighboring locations (Boardman and Kruse, 2011). Since the spatial field was mostly uniform over small distances, these differences conservatively estimated the measurement noise $\sigma$ in each channel. For $n$ cross-track locations, we applied the nonparametric variance estimate of Von Neumann (1941), reprised in Brown and Levine (2007):

$$\hat{\sigma}^2 = \frac{1}{2(n-1)} \sum_{i=1}^{n-1} (\rho_{i+1} - \rho_i)^2 \tag{8}$$

We then characterized the fit using the reduced $\chi^2$ measure (Eldering et al., 2017), a statistical summary of the fit error relative to the expected measurement errors. For $m$ spectral channels, with measured TOA reflectance spectrum $\rho$ and the model estimate $\hat{\rho}$, the $\chi^2$ error was:

$$\chi^2 = \frac{\sum_m (\rho - \hat{\rho})^2}{\sum_m \hat{\sigma}^2} \tag{9}$$





The summations ran over all $m$ spectral channels.

## 2.5 AIRS Comparison

Next, we compared the resulting thermodynamic phase retrievals to a decadal dataset of cloud phase retrievals by the Atmospheric InfraRed Sounder (AIRS) instrument (Pagano et al., 2003). This was a dramatically different measurement obtained

from thermal infrared spectra with a coarse 13.5 km footprint rather than reflected solar energy. We filtered clouds using an AIRS sensitivity threshold (Effective Cloud Fraction, or ECF) of 0.1 (Kahn et al., 2014), and binned AIRS phases by latitude and season for direct comparison.

We anticipated several differences in the result. First, AIRS sampled uniformly over the Earth's surface while the Hyperion dataset favored land areas. We also expected differences in sensitivity; AIRS was far more sensitive to thin clouds, while the

Hyperion analysis intentionally excluded them with a strict cloud mask. For this reason, we normalized the relative cloud abundances for the comparison. We designated a reference latitude band from -60 to 60 degrees (the area of densest Hyperion coverage) to have a mean occurrence of unity, and compared zonal changes relative to this standard.

Additionally, the AIRS algorithm classified ambiguous clouds as "unknown." This population likely contained mixed phase clouds but also a large fraction of supercooled liquid clouds due to the current AIRS phase algorithm. The liquid tests are

based on warm liquid water indices of refraction rather than the supercooled liquid indices of refraction (Rowe et al., 2013). Following from results of Jin and Nasiri (2014), we reassigned some of the unknown clouds to form 10% of total ice and 60% of total liquid. For liquid clouds, we defined a corrected occurrence $L'$ in each latitude bin and took just 40% of this to be from the original AIRS estimate $L$:

$$(1-0.6)\sum L' = \sum L \tag{10}$$

where summations ran over all latitude bins. This allowed a unique correction for each latitude bin, given as the product of its unknown cloud fraction $U$ and a global multiplicative coefficient $\alpha$:

$$L' = L + \alpha U \tag{11}$$

Equations 10 and 11 yield:

$$\alpha = \frac{0.6\sum L}{(1-0.6)\sum U} \tag{12}$$

We defined a similar relation for ice cloud with a missing fraction of 0.1 rather than 0.6. Together, the two adjustments (magnitude normalization and reassignment of unknown clouds) accounted for known biases which permitted a comparison of zonal gradients between the two instruments. This provided a useful check between two very different measurement techniques.

## 2.6 Spatial Scale Analysis

We next characterized spatial scaling properties of the thermodynamic phase maps. A variogram (Garrigues et al., 2006)

estimated the expected squared differences between LTF $x$ at any two locations in the map, written as $v(d)$, a function of the





*lag distance* $d$ between the paired points. We used the classical estimator based on the sample variance at each lag distance:

$$v(d) = \frac{1}{2N(d)} \sum_{\|x-x'\|=h} (x - x')^2 \tag{13}$$

where $N(d)$ was the number of such points in the sample. The näive formulation involving calculations of all point pairs would have required over $10^{10}$ squared differences per scene, an intractable number. Instead, we calculated the same quantity

efficiently in the Fourier domain using the method of Marcotte (1993), with a spatial mask to ensure that only cloud pixels influenced the calculation.

We fit the resulting variogram with a power law of the form $v(d) = ad^b + c$, subsampling the data to achieve log-constant point density and optimizing free parameters with the Levenberg-Marquardt method. We considered the hypothesis that tropical clouds would have different spatial scaling from extra tropical clouds because of the dominant influence of convective versus

baroclinic systems, respectively. To test this, we analyzed the scenes in three segments: a tropical band within 20 degrees of the equator, and extra tropical scenes poleward of 30 degrees North and South latitude. We used large cloud fields beyond a 3 km cutoff, excluding variograms that became degenerate at shorter distances.

## 3   Results

The retrievals clearly revealed distinct cloud phases. Figure 2 shows a typical example subimage. Figure 3 is the corresponding

$EWT_{liquid}$ and $EWT_{ice}$ mapped to green and blue channels, respectively. LTF values in the mixed phase region range from 0.5-0.75, and values in ice areas are typically 0-0.2. Both regions were well-distinguished without significantly overlapping values, but also showed sub-kilometer interior spatial structure. Figure 4 shows normalized histograms of $\chi^2$ scores for the entire scene, calculated independently for liquid and ice clouds. Fits were generally quite good, with $\chi^2$ scores below the conservative noise estimate, $\chi^2 = 1$. In other words, the model fit the spectrum to within the measurement accuracy for these

cloud spectra. For reference, we also show pixels flagged by the cloud detection as clear sky that did not enter later analyses. As expected, clear sky pixels exhibited higher $\chi^2$ scores representing poorer spectrum fits to vapor, ice and liquid spectra.

Figure 5 shows typical spectrum fits from locations in this subimage, with mixed cloud and ice cloud. The top rows show the model fit to spectrum, confirming that the five-parameter model fit the 1.4 - 1.8 $\mu$m interval. The middle rows shows transmittance due to each absorbing component in the model. The continuum component is not shown. The three absorbers

together were sufficient to explain observed spectrum shapes. Finally, the bottom rows show residual error, which is again below the estimated measurement noise level.

### 3.1   Zonal Statistics and AIRS Comparison

Figures 6 and 7 show liquid and ice cloud phase spatial trends across latitudes, partitioned by season, for all 10 years of observations binned in 10 degree increments. Error bars show 95% confidence intervals for the mean. The dataset clearly

resolved key features apparent in records from other sensors like MODIS and CALIPSO (Hirakata et al., 2014; Hu et al., 2010). A band of ice clouds peaked in the Intertropical Convergence Zone (ITCZ) at approximately $5° - 10°$ latitude, and other





**Figure 2.** Typical fragment of a Top of Atmosphere Reflectance image, drawn from Hyperion product ID EO1H2221282005350110KF and displayed in visible red, green, and blue channels.





**Figure 3.** Thermodynamic phase map corresponding to Figure 2, with green indicating Equivalent Liquid Thickness, and blue indicating Equivalent Ice Thickness.



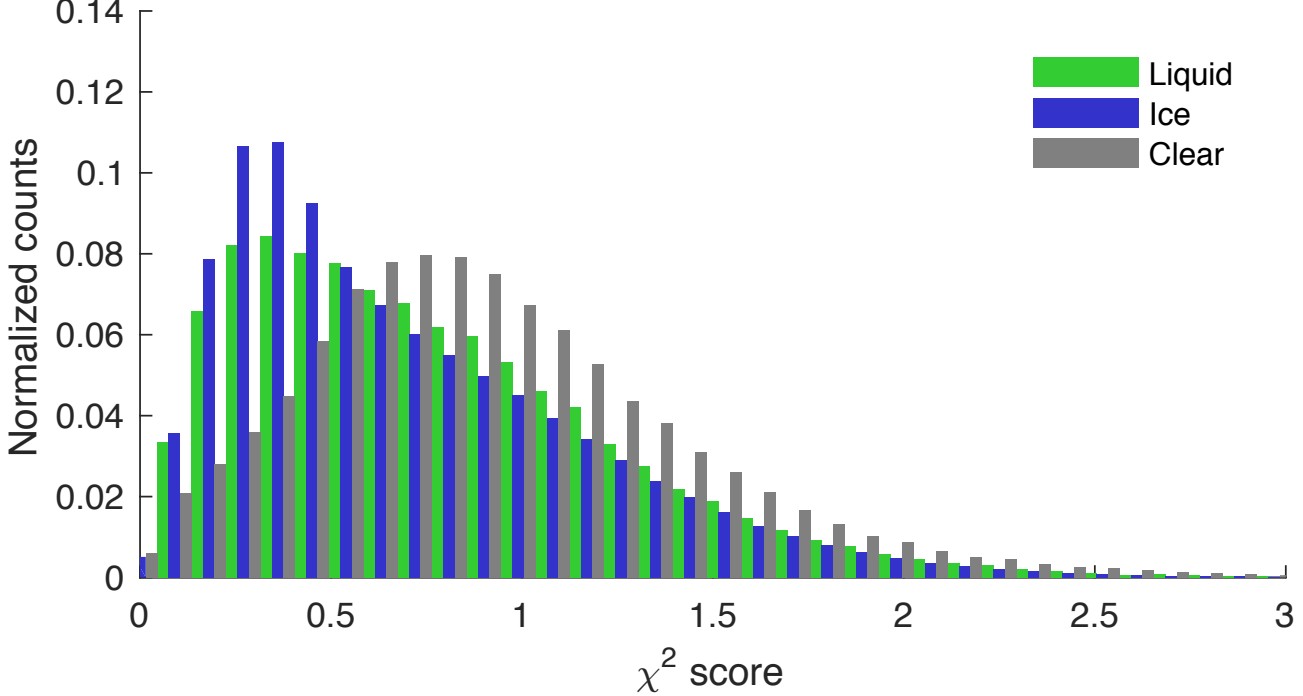

**Figure 4.** Normalized histogram of $\chi^2$ spectrum fit scores for the entire scene of Figures 2 and 3. Low $\chi^2$ values indicate good fits; $\chi^2 = 1$ is a conservative estimate of measurement noise.

seasonally-dependent maxima appeared at approximately 60° and -60°. There was a general increase in the occurrence of liquid clouds at middle and high latitudes, with a strong asymmetry near the solstice: a seasonal peak in liquid cloud coverage for the summer months was apparent in both hemispheres.

Thin lines indicate decadal averages derived from the AIRS dataset. Profiles from the two instruments generally agreed
5  - particularly for ice, to which AIRS was very sensitive. Tropical ice clouds showed the best agreement, as in prior studies comparing AIRS and CALIPSO (Jin and Nasiri, 2014). Differences may be related to the much stronger sensitivity to thin ice cloud, coarse spatial resolution and near global daily sampling, and ambiguity of the unknown category with respect to how many liquid, ice, and mixed phase clouds are contained within that categorization. For reference, we place a non-normalized comparison in Appendix 4.
10  Figure 8 shows the entire distribution of mixed and pure phase clouds, omitting error bars for clarity. Here we show the full range of mixed phase possibilities ranging from pure ice ($LTF < 0.2$) to pure liquid ($LTF > 0.8$). The population associated with the LTF range from 0.6 to 0.8 was fairly large, and possibly included some pure water pixels that were misclassified due to estimation noise. The mixed phase clouds were most numerous in the middle and extreme high latitudes, and nearly absent from the tropics.



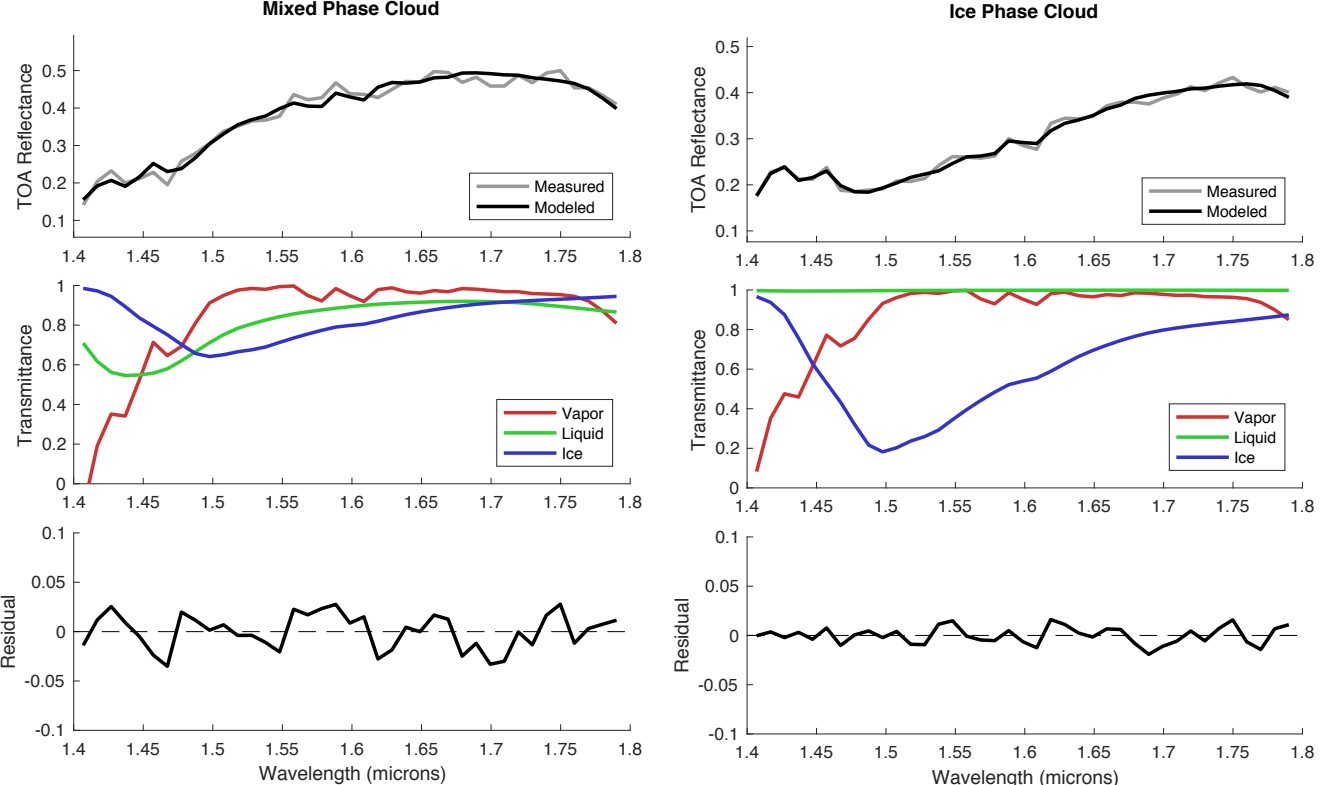

**Figure 5.** Spectrum fits from the region of mixed and ice cloud in Figure 2. Top row: model fit to spectrum. Middle row: Transmittances for each absorber. Bottom row: Residual error.

## 3.2 Spatial Scale Analysis

Finally, Figure 9 shows variograms over all years. The power law fit resulted in the following models for variance in the liquid thickness fraction of tropical clouds, $v_T$, Northern hemisphere extra tropical clouds, $v_N$, and Southern hemisphere extra tropical clouds, $v_S$, as a function of distance $d$ in kilometers:

$$v_T(d) = 0.0026d^{0.62} + 0.0056 \tag{14}$$

$$v_N(d) = 0.0058d^{0.44} + 0.0012 \tag{15}$$

$$v_S(d) = 0.0046d^{0.42} + 0.0010 \tag{16}$$

Differences between the power law exponents for tropical and extra tropical clouds were statistically significant. The fitted exponential scaling factor for tropical clouds was $b = 0.62$ (95% confidence interval of $[0.5998, 0.6398]$). This differed from the exponential factor for extra tropical clouds in the Northern and Southern hemispheres, which were respectively $b = 0.42$ (95% confidence in $[0.3995, 0.4403]$), and $b = 0.44$ (95% confidence in $[0.4147, 0.4679]$). This suggested finer scale variability





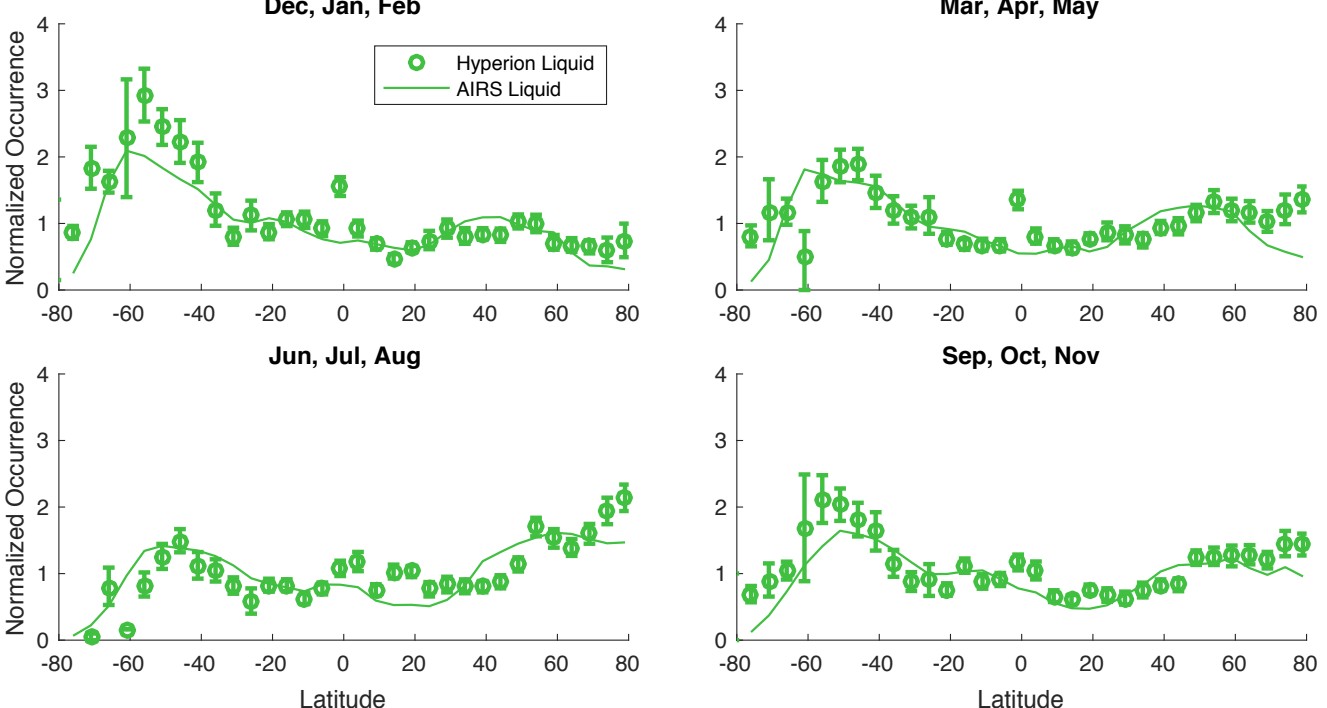

**Figure 6.** Comparison of Hyperion and AIRS cloud statistics, with occurrence normalized by the 0-60 degree latitude range to account for differences in cloud mask cutoff thresholds. The 10-year seasonal observations are shown separately for the liquid phase. Here Hyperion uses a hard classification, i.e., liquid thickness fractions less than 0.5 are considered ice clouds. Error bars show 95% confidence intervals calculated via nonparametric bootstrap estimation.

in cloud thermodynamic phase for clouds outside the tropics. The additive offset terms defined the variance at zero distance, i.e. the irreducible measurement error for each spectrum. The noise equivalent change in liquid water fraction was approximately 7.5% for tropical and 10-11% for extra-tropical clouds. Outside the tropics, this measurement error accounted for just 25% of variance observed at GCM grid scales of 100 km. The remaining 75% was therefore attributable to spatial structure at subgrid

5    scales.

## 4    Conclusions

This study reports the first global high spatial resolution survey of cloud thermodynamic phase. The Hyperion imaging spectrometer provides two novel contributions beyond prior records: first, highly diagnostic spectral features permits accurate discrimination of mixed phase clouds; and second, horizontal scales down to 30 m capture the characteristic spatial scaling

10   relationships of the thermodynamic phase field. Aggregate seasonal and latitudinal changes of cloud thermodynamic phase generally corroborate observations by other sensing modalities, such as those of AIRS. Variogram analysis reveals a noise-



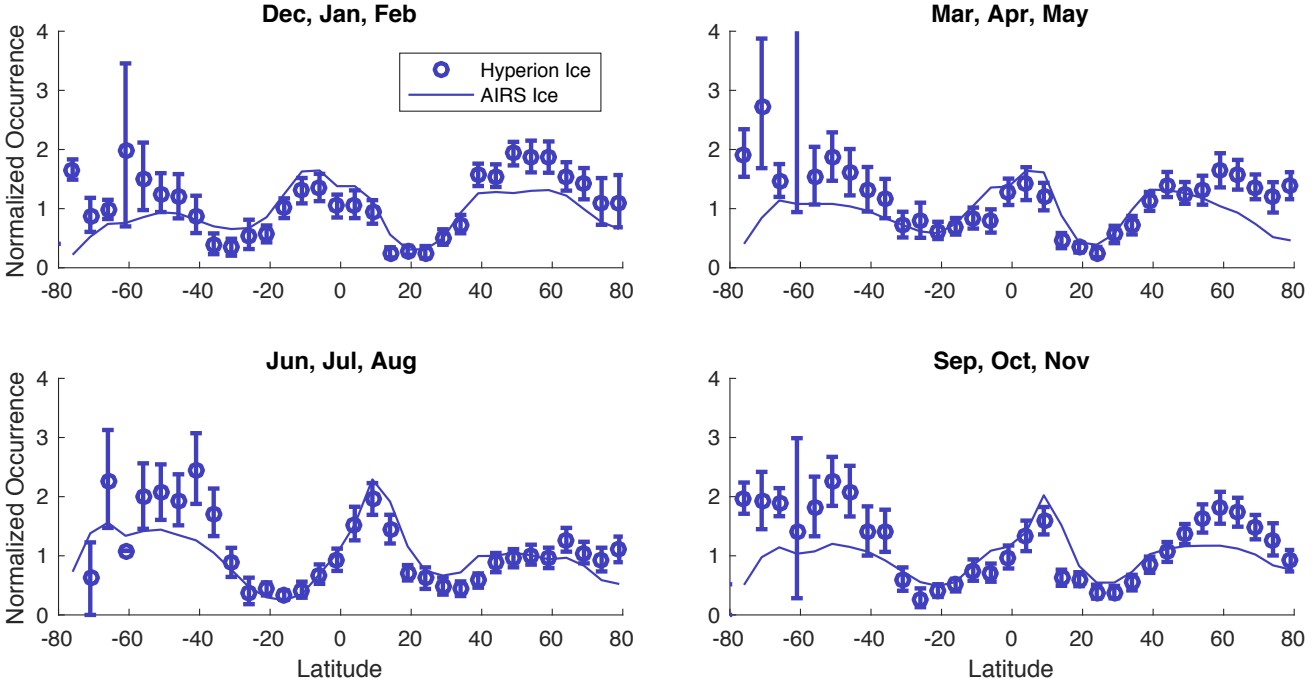

**Figure 7.** Comparison of Hyperion and AIRS cloud statistics as in Figure 6, for the ice phase. Error bars show 95% confidence intervals calculated via nonparametric bootstrap estimation.

equivalent measurement error of 7.5-11% in the liquid thickness fraction for different latitudinal zones. Spatial correlations follow a power law relationship with approximately 50% measurable variance determined at length scales of 6km. Significant spatial variability appears at scales far below the resolution of typical GCMs.

5    We note an important caveat to these results. The Hyperion datasets were spatially biased and strongly favored land mass over ocean. Insofar as the latitudinal trends show asymmetries across northern and southern hemispheres, this may be related to the spatial distribution of land mass in the southern hemisphere midlatitude areas. Southern hemisphere observations were often acquired over islands, which would exhibit a more oceanic influence on cloud cover.

Despite this qualification, the Hyperion results generally corroborate existing global datasets where there is overlap, and provide a "first of a kind" observational record at sub-kilometer scales. These scales are critical for advancing subgrid pa-
10   rameterizations in climate GCMs, including the Wegener-Bergeron-Findeisen time scale of the growth of ice crystals (Tan and Storelvmo, 2016), and numerous other temperature-dependent microphysical processes that control ice and liquid water partitioning (Ceppi et al., 2016), and further extending the observational record to a new extreme of spatial resolution.

*Data availability.* All Hyperion data can be downloaded via http://earthexplorer.usgs.gov





**Figure 8.** Zonal average cloud phase, partitioned by season, for all 10 years of observations.





**Figure 9.** Variogram for all clouds in the Hyperion dataset, showing the variance as a function of separation between points. The curve suggests a power law relationship.




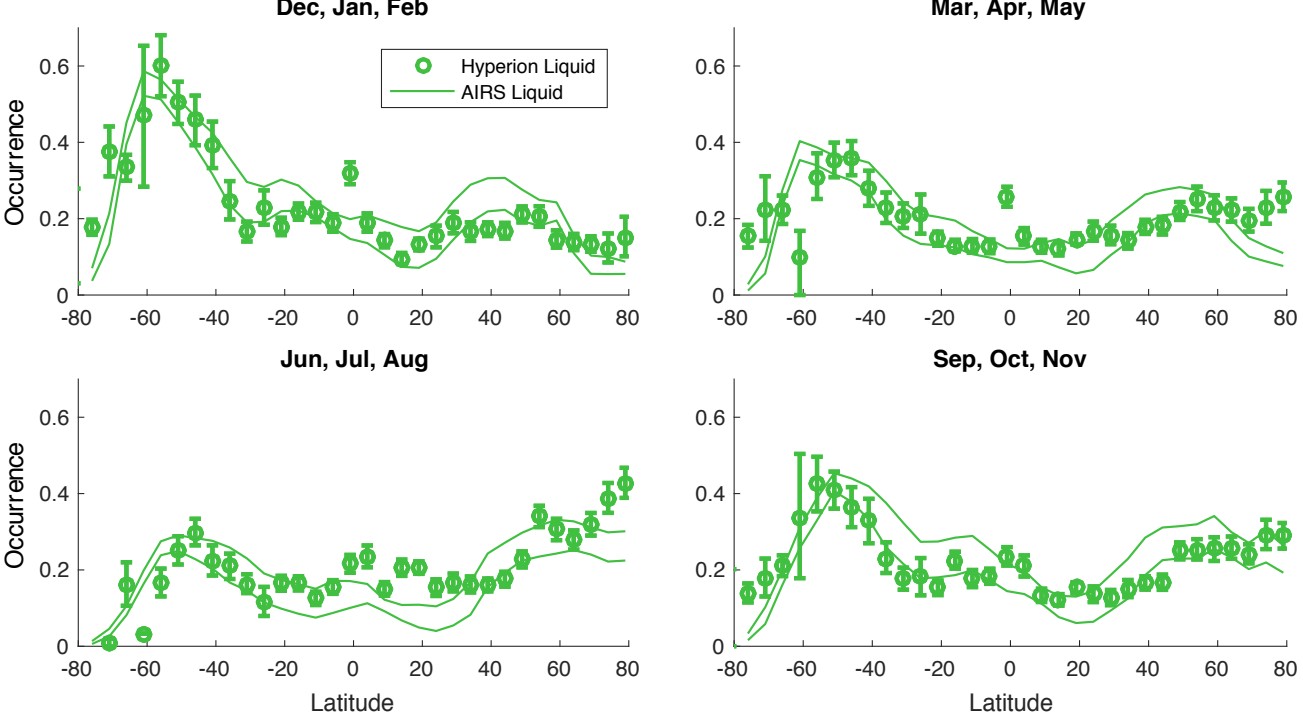

**Figure A1.** A comparison similar to Figure 6 without normalization across latitudes. Two thin lines represent AIRS retrevials using the standard effective cloud fraction (ECF) retrieval threshold of 0.5 and 0.1.

**Appendix A: Non-Normalized AIRS Comparisons**

Figures A1 and A2 show AIRS zonal averages for liquid and ice, respectively, with the correction of Jin and Nasiri (2014) but without normalization across latitudes. Thin lines indicate AIRS frequencies for ECF Thresholds of 0.1 and 0.5. This indicates the sensitivity of the AIRS result to this choice. The absolute abundance of the ice phase shows the largest disparities. This is

5   expected, and accountable to very thin ice clouds to which AIRS is far more sensitive.

*Competing interests.* No author has competing interests.

*Acknowledgements.* This research was performed at the Jet Propulsion Laboratory, California Institute of Technology, under a contract with the National Aeronautics and Space Administration. We thank the Hyperion Team of Goddard Space Flight Center for their assistance in acquiring and interpreting the data. We thank Bo-Cai Gao for radiative transfer methods used in part of the Hyperion analysis. US Government

10  support acknowledged.





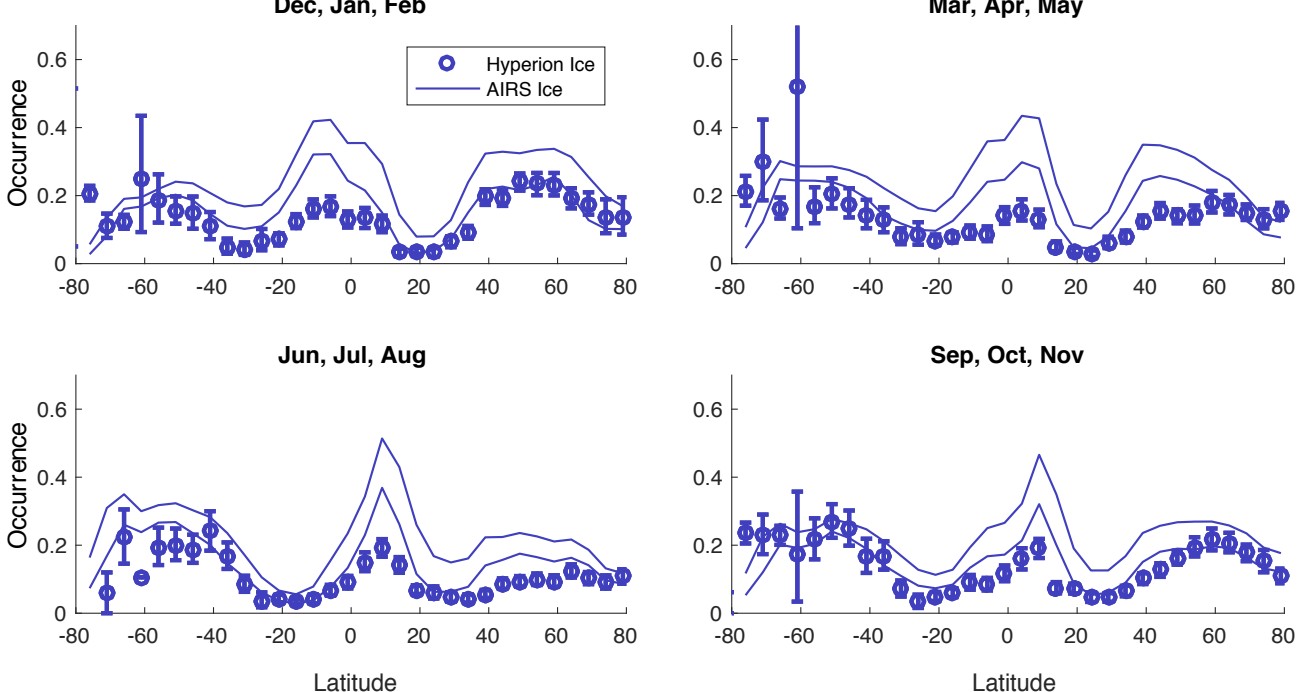

**Figure A2.** A comparison similar to Figure 7 without normalization across latitudes. Two thin lines represent AIRS retreivals using the standard effective cloud fraction (ECF) retrieval threshold of 0.5 and 0.1.

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
