# Peer review of "Global Spectroscopic Survey of Cloud Thermodynamic Phase at High Spatial Resolution, 2005-2015"

_Atmospheric Measurement Techniques, 2017_

## Referee Comment (RC1) · Anonymous Referee #3 · 8 Nov 2017

General Comments:

This is an interesting use of Hyperion/EO-1 data to investigate cloud phase and, broadly speaking, spatial statistics. A significant part of the study is a juxtaposition of Hyperion and AIRS data. Cloud detection and classification methods for both instruments are compared, as are the resulting cloud climatologies for the lifetime of EO-1 (2005-2015).

However, the new and timely result is to use the fine spatial resolution of Hyperion to build scale-by-scale statistics, basically, semi-variograms that show a robust scaling regime once the natural variability emerges from the instrumental noise. A 3-parameter

nonlinear function is used to fit the data. This is a timely development because of the emergence of scale-aware parameterizations of subgrid cloud processes in GCMs.

This manuscript thus has the potential to become a significant contribution to the literature. However, it needs in my opinion a major revision to get there. What made me struggle with the narrative was the decision to use the classic "2. Method" (should be plural) then "3. Results" structure when there are actually three distinct exercises in data analysis: (1) cloud phase classification, (2) comparison with AIRS, and (3) spatial scale analysis.

In their revision, I urge the authors to write three different sections on these topics, each with its subsections on "method" and "results". That way, we don't have to slog through a bunch of method descriptions with applications postponed for several pages. I strongly suggest a "describe, use (show Figs., etc.), and move on" structure iterated three times. That would become a much more powerful build up to the truly new and interesting results on spatial scale analysis.

Questions to address on p. 11, eqs (14-16): Exponent in tropics is $\sim 2/3$, the classic turbulence value. Nice that the N and S extra-tropical counterparts are very close (as are, to some extent the multiplicative and additive constants). But why the significant difference with classic 2/3? Same question about the multiplicative and additive constants? Significance of N-S difference in prefactor of scaling term?

Out of curiosity I plotted the fits in Eqs. (14-16) in 2 ways: same axis limits as in Fig. 9, and lowering the y_min enough to see all. "T" curve looks the same. Something amiss with "N" and "S".

Sequential Comments:

* p. 3, Eq. (1): Use the conventional "/" once, rather than twice "ˆ{-1}".

* p. 3, Eq. (1): Add the customary subscript "_0" to \theta for SZA.

* p. 3, Eq. (2): Coefficient "b" usually precedes the variable (\lambda) in the writing of

a 1st order polynomial.

* p. 3, l. 10: Probably a good idea to list only water vapor, rather than generic gases, since it is the only one considered Then assign it explicitly to j=1, e.g., water vapor (j=1); then repeat, but incremented, for liquid and ice water. Then we know exactly what j is all about.

* p. 3, Eq. (3), line 1: The first 3 terms are used to fit -log(a+b\lambda); only two parameters required, and your (m-n) is one of them. Please make this crystal clear in the math, not just with a vague justification in the following text.

* p. 3, Eq. (3), line 2: - Fuzzy math: a formal vector of parameters can't be $\geq 0$, but each of its elements can. Best to use words in this case. I understand that the fitting algorithm enforces positive values, hence the apparent need to use m and n even though one has to be 0 if the other is not. Better to say that both signs are tested for the "ramp" function.

* p. 4, l. 15: What are the important ETW_x properties? The outcome of (3) maybe? Please define quantity, not just acronym.

* Table 1: I'm a bit confused by Land, Vegetation and Desert ... over ocean, but these are just results from the band math in (5-7). Right? Maybe clarify in caption.

* p. 5, l. 17: Unless ramp slope is redefined earlier, "m" already used.

* p. 6, l. 1: n -> m

* p. 6, l. 30: Do you mean LTF(x)? Rather that denote LTF by x. See next item.

* p. 7, Eq. (13): Isn't h actually d? Also, the squared difference is in _dependent_ variables: LTF(x), I believe.

* p. 7, l. 5: Marcotte (1993) -> (1996), also in References (p. 18, l. 27). Very interesting and, for this study, enabling paper, BTW, that I had to look into. That is how I uncovered the apparent error in year of publication.

* p. 7, l. 12: What is a "degenerate" variogram?

* p. 7, l. 14: This and the next paragraph should be a designated subsection on cloud phase discrimination, or something to that effect, having 3 figures to their credit.

* p. 8, Fig. 2: For completeness, please indicate date and location. And make it visible (different color or arrow?) on Fig. 1.

* p. 10, l. 4: Profiles –> Distributions (since not along vertical here).

* p. 11, l. 9: exponential scaling factor –> scaling exponent

* p. 11, l. 10: exponential factor –> exponent

* p. 13, Fig. 7: Distill caption down to "As in Fig. 6, but for ice phase."

* p. 14, Fig. 8: One legend is enough, preferably located in the middle.

* p. 15, Fig. 9: Why do I see the smallest (maybe Nyquist due to FFT-based estimation?) wavenumber of 90 m? I thought Hyperion pixels were much smaller.

* p. 15, Fig. 9: Please distinguish or mark the "N" and "S" extra-tropical data and fits. Is N/S offset physically significant?

* p. 17, Fig. A2: Distill caption down to "As in Fig. A1, but for ice phase."
* * *
[Figure]

Fig. 1.

[Figure]

**Fig. 2.**

---

## Referee Comment (RC2) · Anonymous Referee #1 · 23 Nov 2017

This manuscript develops a very useful approach to investigate cloud thermodynamic phase based on data from 2005-2015 obtained by the Hyperion imaging spectrometer on EO-1. The approach combines spectrum fitting and spatial scale analysis. The validity is demonstrated with a comparison with AIRS.

I recommend the work for publication in AMT after addressing/clarification of the comments listed below.

Page 3, Eq. (3): The introduction/use of 'm' and 'n' is confusing. The development in equations (2-3) should be clarified. What about error due to this approximation?

Page 4, line 8-9: 'These bulk absorption spectra were generally independent of particle scattering and did not relate directly to particle size'. Is there a reference or evidence for this statement?

Page 5, Eqs. (8-9): Are the "m" and "n" the same variables in Eq. (3)?

Page 5, Eq. (9): $\chi^2$ should be divided by number of degree of freedom because your fit uses a reduced Chi-Square, which is defined as 'Chi-Square per degree of freedom'. So $\chi^2 = 1$ is a conservative estimate of measurement noise as shown in Fig.4. Please clarify.

Page 11, Eqs. (15-16): I believe that the authors made a typo for the offsets. It should be

$$\upsilon N(d) = 0.0058d^{0.44} + 0.012 \qquad (15)$$
$$\upsilon S(d) = 0.0046d^{0.42} + 0.010 \qquad (16)$$

Please check.

---

## Author Comment (AC1) · 27 Nov 2017

color

**1   Reviewer comments**

**Summary**: We agree with all the reviewer suggestions and have incorporated them into a new revision, appended at the end of this document with changes tracked in red. A point-by-point response follows below, with reviewer comments in blue.

[Figure]

This is an interesting use of Hyperion/EO-1 data to investigate cloud phase and, broadly speaking, spatial statistics. A significant part of the study is a juxtaposition of Hyperion and AIRS data. Cloud detection and classification methods for both instruments are compared, as are the resulting cloud climatologies for the lifetime of EO-1 (2005-2015). However, the new and timely result is to use the fine spatial resolution of Hyperion to build scale-by-scale statistics, basically, semi-variograms that show a robust scaling regime once the natural variability emerges from the instrumental noise. A 3-parameter nonlinear function is used to fit the data. This is a timely development because of the emergence of scale-aware parameterizations of subgrid cloud processes in GCMs.

This manuscript thus has the potential to become a significant contribution to the literature. However, it needs in my opinion a major revision to get there. What made me struggle with the narrative was the decision to use the classic "2. Method" (should be plural) then "3. Results" structure when there are actually three distinct exercises in data analysis: (1) cloud phase classification, (2) comparison with AIRS, and (3) spatial scale analysis. In their revision, I urge the authors to write three different sections on these topics, each with its subsections on "method" and "results". That way, we don't have to slog through a bunch of method descriptions with applications postponed for several pages. I strongly suggest a "describe, use (show Figs., etc.), and move on" structure iterated three times. That would become a much more powerful build up to the truly new and interesting results on spatial scale analysis.

We reorganized the manuscript following this suggestion. There are now three independent sections on the cloud phase retrieval, AIRS comparisons, and spatial scale analysis. Each has its own method and results. The content does not change significantly, but we added introduction text to present the organization and improve narrative flow. Overall, we agree with the reviewer on this new organization and feel the manuscript is stronger for the change.

Questions to address on p. 11, eqs (14-16): Exponent in tropics is 2/3, the classic tur-
bulence value. Nice that the N and S extra-tropical counterparts are very close (as are,
to some extent the multiplicative and additive constants). But why the significant differ-
ence with classic 2/3? Same question about the multiplicative and additive constants?
Significance of N-S difference in prefactor of scaling term?

On significance, there are two related questions: (1) is the difference statistically signif-
icant, and (2) is a statistically significant difference physically meaningful? We moved
the model equations into a table of coefficients along with 95% confidence intervals
to show the statistical significance and better isolate the two questions. We have also
added confidence intervals to the curves in Figure 9. The scaling exponents are not
significantly different between Northern and Southern hemispheres; their mutual align-
ment relative to the tropical case demonstrates that even the unevenly-sampled Hyper-
ion dataset shows a significant difference in spatial scaling properties between distinct
cloud populations. We added explanatory text, with two new references:

> We report scaling exponents in terms of variance; these could be trans-
> lated to other conventions using structure functions or the power spectrum
> domain. Differences between the power law exponents for tropical and ex-
> tra tropical clouds were statistically significant. Differences between the
> power law exponents for tropical and extra tropical clouds were statistically
> significant. The extratropical scaling exponents of 0.42 and 0.44 are similar
> to, but slightly in excess of the classic Kolmogorov scaling of 1/3 (-5/3 in the
> power spectral domain). The tropical scaling exponent of 0.62 is in excess
> of the classic Kolmogorov scaling of 1/3 but is consistent with tropical cloud
> reflectance variability reported in Barker et al. (2017), and mid-tropospheric
> water vapor mixing ratio in the tropics from the AIRS instrument, e.g. Kahn
> et al. (2011). At finer spatial resolutions there is also evidence of scale
> breaks dependent on 20 altitude. Consequently the scaling exponent is
> dependent on the length scale range calculated (Kahn et al., 2011).
The additive offset c defined the variance at zero distance, i.e. the irreducible measurement error for each spectrum. This implied a noise equivalent change in liquid water fraction of approximately 7.5% for tropical and 10-11% for extra tropical clouds. The addend and prefactor coefficients differed significantly between extra tropical and tropical clouds, and to a much smaller degree between the two extra tropical populations. We note that all three populations were subject to the Hyperion datasets' biased sampling of ocean and land, and of instrument noise conditions dominated by solar zenith angles. Consequently we would not ascribe the differences between Northern and Southern Hemispheres to cloud scaling properties, since these are small relative to the notable difference with the tropical population. In all cases, measurement error was a small contributor to observed variance — most of the variability arose from spatially-correlated structure. Outside the tropics, measurement error accounted for just 25% of variance observed at GCM grid scales of 100 km. The remaining 75% was therefore attributable to spatial structure at subgrid scales.

Out of curiosity I plotted the fits in Eqs. (14-16) in 2 ways: same axis limits as in Fig. 9, and lowering the $y_min$ enough to see all. T curve looks the same. Something amiss with N and S.

This was due to a typo in our transcription of the offset coefficients, now remedied. The scaling exponents, scaling prefactors, chart, overall trends, and conclusions do not change. We thank the reviewer for catching this error.

**1.1 **Sequential Comments**

- p. 3, Eq. (1): Use the conventional / once, rather than twice "$-1$". Fixed.

- p. 3, Eq. (1): Add the customary subscript $_0$ to $\theta$ for SZA. Fixed.

- p. 3, Eq. (2): Coefficient $b$ usually precedes the variable $\lambda$ in the writing of a 1st order polynomial. Fixed.

- p. 3, l. 10: Probably a good idea to list only water vapor, rather than generic gases, since it is the only one considered Then assign it explicitly to j=1, e.g., water vapor (j=1); then repeat, but incremented, for liquid and ice water. Then we know exactly what j is all about. Fixed.

- p. 3, Eq. (3), line 1: The first 3 terms are used to fit -log(a+b$\lambda$); only two parameters required, and your (m-n) is one of them. Please make this crystal clear in the math, not just with a vague justification in the following text. Fixed.

- p. 3, Eq. (3), line 2: - Fuzzy math: a formal vector of parameters can't be 0, but each of its elements can. Best to use words in this case. I understand that the fitting algorithm enforces positive values, hence the apparent need to use m and n even though one has to be 0 if the other is not. Better to say that both signs are tested for the ramp function. Fixed.

- p. 4, l. 15: What are the important ETW$_x$ properties? The outcome of (3) maybe? Please define quantity, not just acronym. We now define the quantity as the outcome of the prior expression, e.g. the optical absorption path length in millimeters.

- Table 1: I'm a bit confused by Land, Vegetation and Desert ... over ocean, but these are just results from the band math in (5-7). Right? Maybe clarify in caption. We removed these confusing labels.

- p. 5, l. 17: Unless ramp slope is redefined earlier, $m$ already used. Changed to $\ell$ which is unused.

- p. 6, l. 1: n->m We have changed this symbol to $\ell$.

- p. 6, l. 30: Do you mean LTF(x)? Rather that denote LTF by x. See next item. Yes, you are right - we fixed this notation.

- p. 7, Eq. (13): Isn't h actually d? Also, the squared difference is in *dependent* variables: LTF(x), I believe. Fixed; thank you for this catch.

- p. 7, l. 5: Marcotte (1993) -> (1996), also in References (p. 18, l. 27). Very interesting and, for this study, enabling paper, BTW, that I had to look into. That is how I uncovered the apparent error in year of publication. Fixed, thank you.

- p. 7, l. 12: What is a degenerate variogram? We clarified that we excluded scenes with variograms that evaluated to zero at short distances due to the lack of sufficient contiguous cloud pixels.

- p. 7, l. 14: This and the next paragraph should be a designated subsection on cloud phase discrimination, or something to that effect, having 3 figures to their credit. Agreed. we made this change.

- p. 8, Fig. 2: For completeness, please indicate date and location. And make it visible (different color or arrow?) on Fig. 1. Done.

- p. 10, l. 4: Profiles –> Distributions (since not along vertical here). Fixed.

- p. 11, l. 9: exponential scaling factor –> scaling exponent Fixed.

- p. 11, l. 10: exponential factor –> exponent Fixed.

- p. 13, Fig. 7: Distill caption down to "As in Fig. 6, but for ice phase." Fixed.

- p. 14, Fig. 8: One legend is enough, preferably located in the middle. Fixed.

- p. 15, Fig. 9: Why do I see the smallest (maybe Nyquist due to FFT-based estimation?) [wavenumber] of 90 m? I thought Hyperion pixels were much smaller. We assume the reviewer means "lag distance" rather than "wavenumber." We used selected lag distances to ensure a log-constant density. The first two bins corresponded to the identity bin which was not plotted, and 90m. Selected bins were dense near zero, and became widely separated at long ranges.

- p. 15, Fig. 9: Please distinguish or mark the N and S extra-tropical data and fits. Is N/S offset physically significant? The plot has been fixed. See the answer above with respect to significance. We emphasize -and have now written into the text- that each datapoint corresponds to a lag distance for the aggregate variogram, so separation between the curves does not imply separation between populations.

- p. 17, Fig. A2: Distill caption down to "As in Fig. A1, but for ice phase." Fixed.

---

## Author Comment (AC2) · 27 Nov 2017

color

**Summary**: We agree with all the reviewer suggestions and have incorporated them into a new version, appended as a supplement with changes tracked in red. A point-by-point response follows below, with reviewer comments in blue.

This manuscript develops a very useful approach to investigate cloud thermodynamic phase based on data from 2005-2015 obtained by the Hyperion imaging spectrometer

[Figure]

on EO-1. The approach combines spectrum fitting and spatial scale analysis. The validity is demonstrated with a comparison with AIRS. I recommend the work for publication in AMT after addressing/clarification of the comments listed below.

Page 3, Eq. (3): The introduction/use of 'm' and 'n' is confusing. The development in equations (2-3) should be clarified. What about error due to this approximation?

Reviewer 1 made a similar suggestion. We changed this formula to use a single slope value, which makes the linear continuum more obvious. We clarified that we present both positive and negative slope (one of which must be zero) to the nonnegative least squares solver. Naturally, the notion of a continuum is a convenience, and the cloud particle scattering will invariably depart from a perfect linear relationship. However, linear continua over the 1.4-1.8 micron interval were effective in prior modeling and validation experiments (e.g. Thompson et al., JGR 2016). A relatively simple 5 Degree of Freedom model fit all channels within our conservative noise estimate (see discussion below), suggesting that the model captured the major physical processes in play.

Page 4, line 8-9: 'These bulk absorption spectra were generally independent of particle scattering and did not relate directly to particle size'. Is there a reference or evidence for this statement?

This statement was simply a definition of the Kou et al. coefficients. In other words, the attenuation uses the same absorption coefficients that one would use for a thin sheet or volume of water or ice. We have added more text to clarify: "These bulk absorption spectra were molecular properties of $H_2O$, independent of particle size and scattering — a common practice for Shortwave Infrared observations of clouds (Kokhanovsky, 2004)."

Page 5, Eqs. (8-9): Are the "m" and "n" the same variables in Eq. (3)?

Reviewer 1 also noted this repetition; we've switched to a new variable here for clarity.

Page 5, Eq. (9): [Should] be divided by number of degree of freedom because your fit uses a reduced Chi-Square, which is defined as 'Chi-Square per degree of freedom'. So $\chi^2 = 1$ is a conservative estimate of measurement noise as shown in Fig.4. Please clarify.

We added text to clarify: "Specifically, [the reduced $\chi^2$ score was the Chi-square score per degree of freedom, with $\chi^2$=1 equivalent to estimated measurement noise. This was more appropriate than a classical Chi-square test for ur spectroscopic observations where errors could be correlated across adjacent wavelengths."

Page 11, Eqs. (15-16): I believe that the authors made a typo for the offsets. It should be [0.012 and 0.010]. Please check.

Reviewer 1 also noted this. We have corrected the typo in our revision.

Please also note the supplement to this comment:
https://www.atmos-meas-tech-discuss.net/amt-2017-361/amt-2017-361-AC2-supplement.pdf
* * *
[Figure]

**Supplement:**

[revised manuscript text omitted]

20  effective proxy for thermodynamic phase was the Liquid Thickness Fraction (LTF) (Thompson et al., 2016) defined as:

$$\text{LTF} = \frac{\text{EWT}_{liquid}}{\text{EWT}_{liquid} + \text{EWT}_{ice}} \tag{4}$$

We calculated the LTF of locations flagged by the Hyperion cloud detection algorithm of Griffin et al. (2003). In the Griffin et
al. approach, a decision tree of threshold tests sorted spectra into different cloud types (including low and high clouds) as well
as different land cover types (including snow, open terrain surfaces, and vegetation). This favored opaque clouds and generally
25  ignored ambiguous translucent clouds that would be difficult to distinguish from land. The Griffin et al. algorithm defined tests
using the top of atmosphere reflectance $\rho$ and three intermediate quantities, the Normalized Difference Snow Index (NDSI),
the Desert Sand Index (DSI), and the Vegetation Index (VI):

$$\text{NDSI} = [\rho(0.55) - \rho(1.65)] / [\rho(0.55) + \rho(1.65)] \tag{5}$$
$$\text{DSI} = [\rho(0.86) - \rho(1.65)] / [\rho(0.86) + \rho(1.65)] \tag{6}$$
30  $$\text{VI} = \rho(0.66)/\rho(0.86) \tag{7}$$

[revised manuscript text omitted]

---

## Referee Comment (RC3) · Anonymous Referee #4 · 29 Nov 2017

Review of Global Spectroscopic Survey of Cloud Thermodynamic Phase at High Spatial Resolution, 2005-2015, by Thomson et al.

**Recommendation**: this paper has the potential to be an interesting contribution to knowledge, but requires major revision before being accepted as a publication.

**Major comments**

- The comparison with AIRS is flawed. It appears to be a purely statistical comparison, involving mainly land scenes on the part of Hyperion, and global scenes on the part of AIRS. Given the variability of clouds and the sharp differences between maritime and continental clouds, the AIRS data should have been subsetted to match the Hyperion locations. Mention should be made of the differences (or similarity) between the Hyperion and AIRS sampling: I think that they sample at completely different times of the day? So even if both instruments were retrieving cloud phase perfectly, the comparison would be flawed by the different sampling strategies. The results shown in Figs. 6, 7, A1 and A2 consequently are troublesome to interpret. They should have addressed the sampling errors of each instrument, not simply a vague error bar for Hyperion and nothing for AIRS. The comparison is also weakened by the empirical correction factors to AIRS data discussed in section 2.5.
- The definition of LTF is flawed. The signal measured is based on the absorption of solar radiation integrated over the entire photon pathlength, yet eq. 2 refers only to the thickness of the cloud. By their nature, clouds are heterogeneous, so that horizontal variability dominates the radiative transfer process. [This also means that the retrieval technique is at a coarser scale than the postulated 30 m due to the effects of radiative smoothing, and is likely closer to 100 m.] I think this is correctly acknowledged in p.4 line 13 ff. However, it is not really clear whether the LTF is being interpreted correctly. I take it to be the fraction of average photon path that is liquid. Not the fraction of the cloud that is liquid, which would require all paths to extend to the cloud base. An opaque cloud has little transmission, so that most of the reflected paths relate to the top of the cloud. This probably doesn't matter much for the Hyperion retrievals standing alone, but becomes troublesome when compared to other techniques that sample cloud tops differently. It would be good to see a clearer discussion of what is meant by the 'effective proxy for thermodynamic phase'.
- Section 2.3 is a strange, stand-alone paragraph that seems incomplete. How is 'dominant' defined? Greater that 50%? What comparisons were made with historical datasets? This section should be rewritten to provide better context, or incorporated elsewhere.
- Section 2.4 presumably refers to the uncertainty in determining the LTF of a single scene, but this is not clear. It also stops abruptly with no relation to the

results. This needs to be rewritten for clarity and context.

- Section 2.5 should provide a reference to how AIRS obtains cloud phase and whether this has ever been validated.
- The use of the word 'trends' p.5, p7, p.13. This is better reserved for long term climate change. Here we are looking at 'relative dependence on latitude' or similar.
- Fig. 4 shows results for clear retrievals, yet the scene looks completely overcast. Are these all in error, despite the low values of $\chi^2$ for many of these? Given the range of $\chi^2$ shown, presumably the only results retained where when $\chi^2$ was less than some threshold? This could be discussed better.
- Fig.5 is too cryptic for the typical reader. If the vapor transmittance around 1.4 μm is zero, how can there be any reflectance to work with? Does the theory include the vapor paths both above and within the cloud? Probably need to explain what is meant by transmittance in this context.
- Normalization of occurrence: p.6, l.11 is -60 to +60°, Fig. 6 is 0 to 60°. Which is it? Is the normalization done separately for each cloud phase? Are the AIRS data similarly normalized?
- Fig. 8 is flawed by the nonuniform sampling with latitude. Perhaps an indication of the relative number of samples per histogram would help.
- p.10 line 9. Appendix 4? Appendix A.
- Fig. 9 shows NH and SH curves for extra tropical clouds, but which is which? Eq. 14-16 don't seem to match the values on the figure.
- p.13, line4. This caveat comes far too late in my opinion as it dominates the comparison throughout. Note that CALIOP also offers high-resolution phase information that also has fewer sampling limitations.

---

## Author Comment (AC3) · 6 Dec 2017

color

**Summary**: We generally agree with the reviewer suggestions and have incorporated them into a new version, appended as a supplement with changes tracked in red. A point-by-point response follows below, with reviewer comments in blue.

Recommendation: this paper has the potential to be an interesting contribution to knowledge, but requires major revision before being accepted as a publication.

[Figure]

The comparison with AIRS is flawed. It appears to be a purely statistical comparison, involving mainly land scenes on the part of Hyperion, and global scenes on the part of AIRS. Given the variability of clouds and the sharp differences between maritime and continental clouds, the AIRS data should have been subsetted to match the Hyperion locations. Mention should be made of the differences (or similarity) between the Hyperion and AIRS sampling: I think that they sample at completely different times of the day? So even if both instruments were retrieving cloud phase perfectly, the comparison would be flawed by the different sampling strategies. The results shown in Figs. 6, 7, A1 and A2 consequently are troublesome to interpret.

We agree the two datasets are different. We would suggest that this is not necessarily a flaw, but rather conditions one must consider in interpreting the comparison. We intend it mainly as a "sanity check" of broad latitudinal distributions, and feel it is independently interesting for the fact that the instruments use very different measurement strategies. However, we agree that the differences are important and our revision calls them out from the start:

> [This] was a dramatically different measurement obtained from thermal infrared spectra with a coarse 13.5 km footprint rather than reflected solar energy at fine spatial resolution. Kahn et al. (2014) detail the algorithm, and Jin and Nasiri (2014) validate it using pixel-scale comparisons with CALIPSO data [....] We anticipated several differences in the result. First, AIRS sampled uniformly over the Earth's surface while Hyperion imaged only during the day and favored land areas. We also expected differences in sensitivity; AIRS was far more sensitive to thin clouds, while the Hyperion analysis intentionally excluded them with a strict cloud mask [...] Additionally, the AIRS algorithm classified ambiguous clouds as "unknown." This population likely contained mixed phase clouds but also a large fraction of supercooled liquid clouds due to the current AIRS phase algorithm [...] While we expected some discrepancies due to differences in instruments

and sampling, the comparison provided a useful check between two very different measurement techniques.

Regarding the comments on comparing at the pixel scale, we agree with the reviewer that this is the most robust method that minimizes temporal and spatial mismatching. However, since the two instruments are in different orbits, and the Hyperion record is limited to targeted acquisitions, this would leave insufficient coincidences to provide robust statistics. A strict spatial distance and time difference criterion would have to be used in order to account for diurnal variability. These issues would persist even if we subset AIRS to the spatial points of the EO-1 observations, without requiring exact temporal coincidence.

Naturally, it is fairly common practice to compare data sets that retrieve the same geophysical variable and make statistical comparisons from completely different satellite platforms and spatial/temporal sampling. This is done with cloud properties including cloud microphysics, IWP, and LWP. One example is Stubenrauch et al., ASSESSMENT OF GLOBAL CLOUD DATASETS FROM SATELLITES: Project and Database Initiated by the GEWEX Radiation Panel, Bull Amer Met Soc, 2013. These types of comparisons still yield scientifically important insights.

Finally, we would emphaisze that the main Hyperion result stands apart as an independent contribution, and that the latitudinal distributions are consistent those reported for more similar instruments such as MODIS, as summarized in Hirakata et al. (2014). The main contributions of the manuscript are to demonstrate the first global scale cloud phase measurement from reflectance spectroscopy, to provide the first global study imaging cloud phase at 30 m spatial sampling, and to assess spatial scaling properties.

They should have addressed the sampling errors of each instrument, not simply a vague error bar for Hyperion and nothing for AIRS. The comparison is also weakened by the empirical correction factors to AIRS data discussed in section 2.5.

Our revision clearly describes our methodology for calculating 95% confidence intervals: "[We calculated] confidence intervals with nonparametric bootstrap variance estimation (Wasserman, 2006) that resampled the dataset 10,000 times with replacement." We now add that "The corresponding AIRS error bars would be far smaller due to the large number of samples, so we omit them for clarity."

The definition of LTF is flawed. The signal measured is based on the absorption of solar radiation integrated over the entire photon pathlength, yet eq. 2 refers only to the thickness of the cloud.

In fact, the reviewer has interpreted our equation exactly as we had intended: the LTF refers to the absorption along the photon optical path, with no implication for the physical vertical dimensions of the cloud. We call it a "thickness" for consistency with prior literature, such as Gao and Goetz (1995). This is also consistent with our own previous usage (e.g. Thompson et al. 2015, 2016).

By their nature, clouds are heterogeneous, so that horizontal variability dominates the radiative transfer process. [This also means that the retrieval technique is at a coarser scale than the postulated 30 m due to the effects of radiative smoothing, and is likely closer to 100 m.] I think this is correctly acknowledged in p.4 line 13 ff. However, it is not really clear whether the LTF is being interpreted correctly. I take it to be the fraction of average photon path that is liquid. Not the fraction of the cloud that is liquid, which would require all paths to extend to the cloud base. An opaque cloud has little transmission, so that most of the reflected paths relate to the top of the cloud. This probably doesn't matter much for the Hyperion retrievals standing alone, but becomes troublesome when compared to other techniques that sample cloud tops differently. It would be good to see a clearer discussion of what is meant by the "effective proxy for thermodynamic phase."

We modified the text to clarify our definition:

In summary, following Equation 3 we modeled the entire interval from $1.4 - 1.8\ \mu$m with five free parameters: a continuum offset $l$; a slope, represented by a single degree of freedom in the variables $m$ and $n$; and the vapor, ice and liquid thicknesses $u_j$. These thicknesses represented the length of the optical path through an equivalent homogeneous volume, as in the Equivalent Water Thickness (Gao and Goetz, 1995). As in previous work, we wrote the absorption path length $u_2$ as the Equivalent Water Thickness due to Liquid in millimeters, $\text{EWT}_{liquid}$. Similarly, $u_3$ was the Equivalent Water Thickness due to ice, $\text{EWT}_{ice}$. We then defined the Liquid Thickness Fraction (LTF) as:

$$\text{LTF} = \frac{\text{EWT}_{liquid}}{\text{EWT}_{liquid} + \text{EWT}_{ice}} \tag{1}$$

Prior in situ validation had demonstrated a robust relationship between the LTF and thermodynamic phase (Thompson et al., 2016). We emphasize that "thickness" referred to the absorption along the optical path; clouds were heterogeneous, so the LTF was not necessarily related to their vertical dimension. In opaque clouds the measurement would be most sensitive to the upper layers.

Section 2.3 is a strange, stand-alone paragraph that seems incomplete. How is 'dominant' defined? Greater that 50%? What comparisons were made with historical datasets? This section should be rewritten to provide better context, or incorporated elsewhere.

Following a comment by another reviewer, we have restructured the manuscript to follow a more thematic organization which places this paragraph in better context. We have modified the text to indicate that we used a 50% cutoff threshold.

Section 2.4 presumably refers to the uncertainty in determining the LTF of a single scene, but this is not clear. It also stops abruptly with no relation to the esults. This

needs to be rewritten for clarity and context.

We agree; our manuscript restructuring clarifies the implication of these $\chi^2$ values. The fits demonstrate that the model explains the variability observed in the spectra to within our noise estimate, showing that the retrieval method of Thompson et al. (2016) also applies to Hyperion. We have added text to this effect. We also state that noise is calculated on a per-line basis (it is dominated by constant factors like the solar zenith). However, we calculate $\chi^2$ statistics independently for each spectrum, since we fit a model independently for each 30 m × 30 m spatial location.

Section 2.5 should provide a reference to how AIRS obtains cloud phase and whether this has ever been validated.

We now state: "This was a dramatically different measurement obtained from thermal infrared spectra with a coarse 13.5 km footprint rather than reflected solar energy at fine spatial resolution. Kahn et al. (2014) detail the algorithm and and Jin and Nasiri (2014) validate it using pixel-scale comparisons with CALIPSO data."

The use of the word 'trends' p.5, p7, p.13. This is better reserved for long term climate change. Here we are looking at 'relative dependence on latitude' or similar.

We substitued the term "distributions."

Fig. 4 shows results for clear retrievals, yet the scene looks completely overcast. Are these all in error, despite the low values of $\chi^2$ for many of these? Given the range of $\chi^2$ shown, presumably the only results retained where when $\chi^2$ was less than some threshold? This could be discussed better.

We understand how the clear sky case could be confusing. In fact, the scene is a fragment of a larger image that included clear sky areas, and these statistics come from the clear parts (which are outside the area shown in our figure). Ironically, the $\chi^2$ values for the clear cases had been low despite the fact that the instrument saw the Earth's surface. The image was acquired over an ice shelf and ocean, and the former

showed ice absorption while the latter had nearly zero reflectance in this range. Since we do not include clear sky cases in our global statistics (for obvious reasons) there is no reason to report these $\chi^2$ values. For clarity, we have removed the case from the histogram and the text.

Fig.5 is too cryptic for the typical reader. If the vapor transmittance around 1.4 $\mu m$ is zero, how can there be any reflectance to work with? Does the theory include the vapor paths both above and within the cloud? Probably need to explain what is meant by transmittance in this context.

Thank you - this was a good catch. It was an artifact of our figure, which had used the first order Taylor approximation of the transmittance. The approximation was not accurate near saturation. We replotted the figure using the true transmittances, which were naturally greater than zero. The product of plotted transmittances (and the continuum, not shown) now reproduces the observed TOA reflectance. The changes are very subtle outside the 1.4 micron vapor feature, and the general shapes and relative depths are not significantly altered.

Normalization of occurrence: p.6, l.11 is -60 to +60°, Fig. 6 is 0 to 60°. Which is it? Is the normalization done separately for each cloud phase? Are the AIRS data similarly normalized?

This was a typo in the figure caption; the normalization uses the whole -60 to +60° interval. We updated the caption, and changed the text to emphasize that both AIRS and Hyperion are normalized.

Fig. 8 is flawed by the nonuniform sampling with latitude. Perhaps an indication of the relative number of samples per histogram would help.

We have added a note to the caption and text reminding the reader that sampling is nonuniform, and that the results should be evaluated in light of the bootstrap uncertainty analysis. This figure is presented in the context of Figures 6-7, where bootstrap

variance estimation shows the uncertainty due to sample size in each histogram bin.

p.10 line 9. Appendix 4? Appendix A.

Fixed, thank you.

Fig. 9 shows NH and SH curves for extra tropical clouds, but which is which? Eq. 14-16 don't seem to match the values on the figure.

We have remedied a typo in the offset values for these equations - now they match the figure. We have also changed the figure, labeling the two extra tropical curves.

p.13, line 4. This caveat comes far too late in my opinion as it dominates the comparison throughout. Note that CALIOP also offers high-resolution phase information that also has fewer sampling limitations.

The new version emphasizes nonuniformity throughout, in the following locations:

- Section 2.1: "[Hyperion] performed targeted acquisitions for specific regions of interest, with occasional pointing off nadir. Most targets were on land, with a high concentration in the mid-latitude northern hemisphere. There was sparser coverage of extreme latitudes and oceans, but several island targets offered a view into cloud systems over ocean (Figure 1). Many targets of interest were revisited multiple times during the mission."

- Figure 1: portrays the Hyperion image locations

- Section 2.2: "Note that Hyperion sampling is nonuniform across 20 histogram bins, and Section 1 quantifies uncertainty for different latitudes."

- Section 3.1: "AIRS sampled uniformly over the Earth's surface while the Hyperion dataset favored land areas."

- Section 5 (conclusions): "The Hyperion datasets were spatially biased and strongly favored land mass over ocean. Insofar as the latitudinal trends show

asymmetries across northern and southern hemispheres, this may be related 15
to the spatial distribution of land mass in the southern hemisphere midlatitude ar-
eas. Southern hemisphere observations were often acquired over islands, which
would exhibit a more oceanic influence on cloud cover."

We believe that the closing discussion is an appropriate place to contextualize these
results and draw implications. Obviously the Hyperion mission was not designed for
cloud observations. However, the ability to form cloud phase maps at 30m resolution,
is a unique new capability that makes the investigation meritorious. Additionally, results
generally agree with existing cloud phase records from AIRS (and multi-instrument
comparisons such as Hirakata et al., 2014). Because the measurement technique is
distinct, is is a useful complement to other instruments using polarization (CALIOP) or
thermal emission (AIRS). We have modified the conclusion to emphasize this. Finally,
the Hyperion datasets provide a "first of a kind" observational record at sub-kilometer
scales. Spatial granularity reaches a factor of three below CALIOP observations,
though one should also note the reviewer's caveat about within-cloud scattering
placing a lower limit on achievable resolution.

Please also note the supplement to this comment:
https://www.atmos-meas-tech-discuss.net/amt-2017-361/amt-2017-361-AC3-
supplement.pdf

---

## Referee Comment (RC4) · Anonymous Referee #2 · 19 Dec 2017

General comments This paper presents the first global study of cloud phase derived from shortwave infrared (SWIR) reflectance spectra at 30 m spatial sampling and spatial scaling properties of cloud phase. I found this paper interesting. The manuscript has been already revised based on previous reviewers' comments. I have only a few questions and comments. When I am referring page and figure numbers below, I am referring the latest version of revised manuscript (AC3 supplement).

1. Multilayered cloud systems: I found no description on how multilayered cloud systems are detected and handled in this study. In my view, "ice" cloud region shown in Fig. 2 looks like a multilayered cloud system with an optically thin, high cloud above

an optically thick, low cloud deck. I am not sure on this because I am not an expert of this kind of imagery, but I was wondering why "ice" cloud region is more reflective than "mixed phase" cloud region. Satellite measurements show that multilayered cloud systems are quite common in the tropics and mid-latitude storm track regions. Thermal infrared measurements by AIRS are sensitive to the upper cloud, but the SWIR reflectance from Hyperion should be more sensitive to the lower cloud, depending on the optical thickness of upper cloud. If so, there should be more liquid cloud occurrence in Hyperion's results than in AIRS, in specific latitude zones. Is this a possible reason for statistically significant Hyperion–AIRS differences in the tropics and mid-latitude storm track regions, as in Figs 7 and 8? The authors just mentioned that distributions from the two instruments generally agreed and the differences were ascribed to sampling error and spectroscopic sensitivity difference. In my opinion, if there is a statistically significant difference, that difference is valuable to be discussed and should be clarified in the manuscript. In that way, this comparison is not just a "sanity check" but more valuable.

2. On the comparison with AIRS: Oceanic and continental averages of cloud phase fraction can be derived from AIRS data. How can the difference between them explain the difference between results from the Hyperion and AIRS? It would be more insightful to compare the Hyperion's results with AIRS oceanic and continental averages.

Specific comments Page 6, Line 30, "The mixed phase clouds were ... nearly absent from the tropics": It seems to be not nearly absent.

Page 9, line 15, "thin cloud": Is this an optically thin, high (or low) cloud?

---

## Author Comment (AC4) · 22 Dec 2017

**Summary**: We generally agree with the reviewer suggestions and have incorporated them into a new version, appended as a supplement with changes tracked in red. A point-by-point response follows below, with reviewer comments in blue.

Multilayered cloud systems: I found no description on how multilayered cloud systems are detected and handled in this study. In my view, "ice" cloud region shown in Fig. 2 looks like a multilayered cloud system with an optically thin, high cloud above an optically thick, low cloud deck. I am not sure on this because I am not an expert of

[Figure]

this kind of imagery, but I was wondering why "ice" cloud region is more reflective than "mixed phase" cloud region.

In fact, the reflectance of the mixed phase is slightly higher than for the ice phase. This is captured in our new plot. It is indeed possible that the ice cloud hides a liquid cloud below (see below).

Satellite measurements show that multilayered cloud systems are quite common in the tropics and mid-latitude storm track regions. Thermal infrared measurements by AIRS are sensitive to the upper cloud, but the SWIR reflectance from Hyperion should be more sensitive to the lower cloud, depending on the optical thickness of upper cloud. If so, there should be more liquid cloud occurrence in Hyperion's results than in AIRS, in specific latitude zones. Is this a possible reason for statistically significant Hyperion–AIRS differences in the tropics and mid-latitude storm track regions, as in Figs 7 and 8? The authors just mentioned that distributions from the two instruments generally agreed and the differences were ascribed to sampling error and spectroscopic sensitivity difference. In my opinion, if there is a statistically significant difference, that difference is valuable to be discussed and should be clarified in the manuscript. In that way, this comparison is not just a "sanity check" but more valuable.

This is an excellent point - to the degree that there is very optically thin ice cloud above a liquid cloud, AIRS and Hyperion might give two different answers. We agree that this is one potential explanation for differences in AIRS and Hyperion, in addition to spatiotemporal sampling. More generally, AIRS and Hyperion will be sensitive to different altitudes within a large cloud. We have added a note to this effect in the AIRS section, and with due deference to the reviewer, will incorporate this phraseology directly: "Another potential contributor to the discrepancy is sensitivity to different altitudes in large or multilayer cloud systems. Multilayered clouds are abundant in tropical regions and and mid-latitude storm tracks. In cases where, for example, a translucent ice cloud overlays an optically thin liquid cloud, the two instruments would measure different thermodynamic phase."

On the comparison with AIRS: Oceanic and continental averages of cloud phase fraction can be derived from AIRS data. How can the difference between them explain the difference between results from the Hyperion and AIRS? It would be more insightful to compare the Hyperion's results with AIRS oceanic and continental averages.

We absolutely agree that the next natural step in a comparison would be a closer comparison of specific spatiotemporal subsets. It was not obvious how to do that in a paper of this scope, since there is insufficient direct coincidence to provide strong statistics, and there are other differences beyond the continent/ocean biases - for example, the fact that Hyperion observed only during the day, with observations concentrated in areas with human populations. Our response to reviewer 3 describes some of the other differences. We felt that the current evaluation was a simple story, and that a partial remedy of sampling differences might mislead the readership into expectations of precise alignment.

Page 6, Line 30, "The mixed phase clouds were ... nearly absent from the tropics": It seems to be not nearly absent.

Agreed; we changed "nearly absent" to "less abundant."

Page 9, line 15, "thin cloud": Is this an optically thin, high (or low) cloud?

[revised manuscript text omitted]